# Effects of Extrusion on Mechanical and Corrosion Resistance Properties of Biomedical Mg-Zn-Nd-xCa Alloys

**DOI:** 10.3390/ma12071049

**Published:** 2019-03-30

**Authors:** Gui Lou, Shumin Xu, Xinying Teng, Zhijian Ye, Peng Jia, Hao Wu, Jinfeng Leng, Min Zuo

**Affiliations:** School of Materials Science and Engineering, University of Jinan, Jinan 250022, China; lougui18@126.com (G.L.); 18764415128@163.com (S.X.); yezj8889@163.com (Z.Y.); jiapeng@mail.ujn.edu.cn (P.J.); mse_wuh@ujn.edu.cn (H.W.); jfleng@126.com (J.L.); mse_zuom@ujn.edu.cn (M.Z.)

**Keywords:** magnesium alloys, biocompatibility, extrusion, microstructure, mechanical property, corrosion resistance

## Abstract

Magnesium alloys act as ideal biomedical materials with good biocompatibility. In this paper, the extruded biomedical Mg-6Zn-0.5Nd-0.5/0.8Ca alloys were prepared and their microstructure, mechanical properties and corrosion properties were investigated. The results showed that the surfaces of Mg-6Zn-0.5Nd-0.5/0.8Ca alloys extruded at medium temperature were smooth and compact without cracks. The tensile strength and elongation of Mg-6Zn-0.5Nd-0.5/0.8Ca alloys were 222.5 MPa and 20.2%, and 287.2 MPa and 18.4%, respectively. A large number of dislocations were generated in the grains and on grain boundaries after the extrusion. The alloy was immersed in simulating body fluid (SBF) for the weightlessness corrosion, and the corrosion products were analyzed by FTIR, SEM equipped with EDS. It was found that the corrosion rate of Mg-6Zn-0.5Nd-0.5Ca and Mg-6Zn-0.5Nd-0.8Ca alloy were 0.82 and 2.98 mm/a, respectively. Furthermore, the compact layer was formed on the surface of the alloy, which can effectively hinder the permeation of Cl^−^ and significantly improve the corrosion resistance of magnesium alloys.

## 1. Introduction

With the development of society and the improvement of living standards, biomaterials have attracted immense attention [1]. Where the “biomaterial” is man-made biocompatible materials used to replace or assist human organs [2]. Metal materials are promising prospects in the biomedical area, with high strength and good plasticity [3]. At present, there are a wide range of applications in clinical applications such as titanium alloys [4], cobalt chromium alloys [5], and stainless steel materials [6]. However, when these metal materials are in service in vivo, they will produce toxic substances due to the degradation and abrasion of body fluids, which will not only reduce biocompatibility but also cause inflammation [7,8]. Moreover, the elastic modulus of some metal materials is different from that of biological bone tissue, and the stress shielding effect is obvious and the stability is poor, which is not conducive to being used as biological materials [9,10].

Compared with titanium alloys and cobalt-chromium alloys and stainless steel materials, magnesium alloys possess many desirable properties in clinical applications such as lower cost, good biocompatibility, nontoxicity, biodegradability and similar elastic modulus to bone tissue. The biocompatibility, nontoxicity, and biodegradability of magnesium alloys are conducive to the excretion of degradation products through the body’s metabolic process without toxic side effects and secondary surgery [11]. The similar elastic modulus can effectively prevent stress shielding effects [12,13]. Therefore, magnesium alloys become a research hotspot in the field of implantable biomedical materials. Nevertheless, magnesium and its alloys are highly susceptible to corrosion and have low strength and plasticity, which limit their wide application [14,15]. It is well known that alloying strategy is an effective strategy to improve the comprehensive properties of materials [16]. Both calcium and zinc play an important role in alloying, and they are essential metal elements. In addition, the rare earth (RE) elements can also improve mechanical properties and corrosion resistance [17]. Furthermore, the recent report claimed that a small amount of neodymium (Nd) was not toxic to humans [18]. Additionally, the plastic deformation and heat treatment can improve the mechanical properties of magnesium alloys [19,20]. The plastic deformation did not make pure magnesium alloys toxic [21]. Based on the above facts and our previous research [22], the Mg-Zn-Nd-Ca alloys, plastic deformation and heat treatment were selected to further improve the mechanical properties and corrosion resistance of magnesium alloys.

## 2. Experimental Method

### 2.1. Preparation of Investigated Alloys

The designed Mg-6Zn-0.5Nd-0.5/0.8Ca alloys were prepared from pure Mg (99.95 wt%) and Zn (99.98 wt%), Mg-24%Nd, and Mg-22%Ca master alloys. The specific process was as follows. Firstly, the pure Mg and two master alloys were placed in a graphite-clay crucible under the protection of a mixed gas atmosphere of SF_6_ (2%, volume fraction) and CO_2_. Secondly, the pure Zn was added into the melt at 740 °C and the melts were held at 720 °C for 30 min. Finally, the melt was cast into a preheated steel mold. After the preparation, the cast ingots were solution treated at 380 °C for 8 h and subsequently were quenched with the water at 60 °C.

### 2.2. Extrusion and Heat Treatment

According to the previous research [23], the extrusion temperature was divided into two levels: medium and high temperature zone. The temperature ranges are from 200–300 °C and from 300–400 °C, respectively. Before the extrusion, the preheating temperature was carried out at 300 °C. During the extrusion, the extrusion temperature was in the range from 220–280 °C, the extrusion speed was 8 m/min, and the extrusion ratio was 16:1. Figure 1 displays the schematic diagram of extrusion process. After the extrusion, the extruded specimens were aged at 200 °C for 12 h. In addition, the chemical compositions of the extruded alloys were analyzed by X-ray fluorescence spectrometric (XRF) and the results were listed in Table 1. 

### 2.3. Structural Characterization

After the aging treatment, the microstructure and composition of investigated alloys were analyzed by a field-emission scanning electron microscope (SEM, FEI-QUANTA FEG250, Hillsboro, OR, USA) equipped with an energy dispersive spectrometer (EDS). Before the SEM observation, the specimen was processed as follows. Firstly, the inlaid sample successively was ground with sandpaper of 100#, 600#, and 2000#. Secondly, the specimen was polished to a scratch-free mirror using ca. 1 μm alumina polishing solution. Finally, the specimen was etched with 4% nitric acid (volume fraction) for a few seconds and subsequently rinsed several times with the deionized water. Considering that the magnesium alloys were easily oxidized in the air, the SEM observation should be conducted as soon as possible. The dislocation density and phase structure were determined with a transmission electron microscopy (TEM, JEM-2100, JEOL, Tokyo, Japan) under an accelerating voltage of 200 kV. The TEM-sample was prepared as follows. Firstly, the thickness of the sample was ground to ca. 120 μm. Secondly, the sample was cut into wafers with a diameter of 3 mm. Thirdly, the thickness was finely ground to ca. 50 μm. Finally, the sample was thinned by ion-beam thinning technology.

### 2.4. Performance Characterization

The tensile test was performed on a WDW-100A universal testing machine (Sida, Jinan, China). The detailed experimental details can refer to our previous works [24,25]. The corrosion resistance was determined by the electrochemical test and immersion test in simulating body fluid (SBF) at 37 ± 0.3 °C. The concentrations of various ions in the SBF were listed in Table 2. The specimens were cut into a cuboid with the size of 15 × 15 × 5 mm^3^ for the corrosion tests. The electrochemical test was performed on an electrochemical system (Gaoss Union EC500, Zahner, Kronach, Germany). The detailed experimental details can refer to the reference [26,27]. The immersion times were 1, 8, 16, 24, 36, 48, 72, 120, 192, and 288 h, respectively. The pH value was measured by the pH meter during the immersion test. After immersion, the sample was successively cleaned by a mixture solution of CrO_3_ and AgNO_3_ (the mass ratio was 16:1) and acetone solution, then dried and weighed. The corrosion rate was calculated according to Equation (1) [28,29]:(1)$$CR=\frac{8.76\times 10^{4}\times \Delta w}{A\times \rho \times t}$$

Herein, *CR* is the corrosion rate (mm/a). *A* is the exposed area of the alloys (cm^2^). *ρ* is the density of alloy (g/cm^3^). *t* is the immersion time (h).

## 3. Results and Discussion

### 3.1. Appearance of the Extruded Alloys

The extrusion temperature had a significant effect on the properties of the alloy, such as the appearance, mechanical properties and corrosion resistance [30]. In order to study the effect of extrusion temperature on the properties of the alloy, the alloy was extruded at medium temperature and high temperature, respectively. Figure 2a,b showed the appearance of the as-extruded Mg-6Zn-0.5Nd-0.5/0.8Ca alloys at high temperature and medium temperature, respectively. At high extrusion temperature, the extruded Mg-6Zn-0.5Nd-0.5/0.8Ca alloys were severely oxidized and the extruded bars cracked severely. At medium extrusion temperature, the surface of Mg-6Zn-0.5Nd-0.5Ca alloy was smooth and crack-free, while the surface of Mg-6Zn-0.5Nd-0.8Ca alloy emerged squamous structure due to the increase of additional stress.

### 3.2. Microstructure of the Extruded Alloys

Figure 3a showed the SEM image of extruded Mg-6Zn-0.5Nd-0.5Ca alloy. The extruded Mg-6Zn-0.5Nd-0.5Ca alloy was constituted of α-Mg matrix and secondary phase particles with the size from 0.8 to 12.4 μm. The EDS spectra of points 1, 2, and 3 in Figure 3a were shown in Figure 3b, Figure 3c,d, respectively. The matrix phase mainly contained 98.17 at% Mg and 1.83 at% Zn, suggesting that the matrix phase was α-Mg phase. The white phases were composed of Mg, Zn, Nd, and Ca, indicated that the Nd and Ca elements were mainly in the two precipitated phases (namely the Ca-rich and Nd-rich ones). The precipitation phase was spherical at point 1, and was strip at point 2.

The microstructure, mechanical properties and corrosion resistance of alloys extruded at medium temperature were investigated. The microstructure of the as-extruded Mg-6Zn-0.5Nd-0.5Ca alloy was shown in Figure 4a,c. The flaky secondary phase was distributed evenly in the matrix, and the shapes of secondary phases were spherical and strip. The microstructure of the as-extruded Mg-6Zn-0.5Nd-0.8Ca alloy was shown in Figure 4b. The secondary phase was mainly on the grain boundary. In addition, it can be found from Figure 4d that the needle-like phase and point-like secondary phase existed in α-Mg grains.

### 3.3. Mechanical Properties and Strengthening Mechanism

Based on the previous study, the compression mechanical properties of Mg–Zn–Nd–Ca alloys were shown in Figure 5. Herein, the casting, heat treatment alloys and extruded alloys were employed, moreover, ST is the abbreviation of “Solution Treatment”. For the extruded alloys, the fracture strength and yield strength were increased. Compared with extruded Mg–6Zn–0.5Nd–0.5Ca alloy, the fracture strength of extruded Mg–6Zn–0.5Nd–1Ca alloy increased more significantly. 

Stress-strain curves of alloy after the aging treatment were shown in Figure 6 and the mechanical properties of the alloy results were listed in Table 3. The stress rose linearly with the increase of strain in elastic deformation zone. Subsequently, the stress decreased slightly during the yield. Afterwards, the stress rose slowly in the non-elastic deformation zone. Finally, the stress suddenly dropped once the fracture occurred. The elastic limit (σ_e_), yield strength (σ_s_) and tensile strength (σ_b_) of Mg-6Zn-0.5Nd-0.8Ca alloy were higher than those of Mg-6Zn-0.5Nd-0.5Ca alloy, while the elongation was opposite. Hence, the strengthening mechanism was the deformation strengthening, which can be further determined based on the TEM images.

It was known that the magnesium alloy had the *hcp* lattice structure, with few slip systems, and was inferior in strength and plasticity. After extrusion and aging treatment, the microstructure of the alloy was uniform; the grain size was fine, as well as the precipitation phase was evenly distributed on the matrix. From the TEM analysis, a large number of high-density dislocations were present inside the grains. The yield strength and tensile strength of the alloy increased due to dislocation entanglement, intersection and pinning of solid solution atoms.

The TEM analysis was performed to investigate the small particle phase of the Mg-6Zn-0.5Nd-0.8Ca alloy after the aging treatment. According to the bright-field micrograph of Mg-6Zn-0.5Nd-0.8Ca alloy (Figure 7a), a large number of dislocations were in the equiaxed grain and piled up and netted at the grain boundary. In addition, a large number of small particle phases can be found in the α-Mg grains. This was because that the solution treatment makes the secondary phases dissolve into the matrix and the secondary phases re-precipitated after the aging treatment. Figure 7b showed that the dislocations and secondary phase particles interacted with each other through bypassing or entangling mechanisms.

The intersection area of the dislocation line and the secondary phase particles was enlarged and shown in Figure 7c. The dislocation ring generated by dislocation motion was in the circle and the dislocation lines were marked by arrows. After the dislocation line passed through the second phase particle, a dislocation loop was formed around the nanoparticle. According to the Orowan mechanism [31], when the moving dislocation line met the second phase particle, its motion would be blocked and the dislocation line would bend around the particle. With the increase of the extrusion stress, the bending of the dislocation line would intensify. Finally, the dislocation line around the particle met and annihilated each other at the meeting point and formed a dislocation ring surrounding the particle. The rest of the dislocation line would continue to move. The second-phase particles were analyzed by selected area electron diffraction in Figure 7d. The electron diffraction pattern of the secondary phase particles taking along the [0001] crystallographic axis indicated that the phase was a precipitated on the {11$\bar{2}$0} prismatic planes and indexed as MgZn_2_. The lattice constants are a = b = 0.5221 nm, c = 0.8567 nm, α = β = 90°, γ = 120°, and the total number of atoms in the unit cell is 12. Related studies have shown that MgZn_2_ phase was a common Mg-Zn strengthening phase, which played an important role in improving the mechanical properties of magnesium alloys [32].

### 3.4. Corrosion Resistance and Corrosion Mechanism

Figure 8a showed the corrosion rate of the extruded Mg-6Zn-0.5Nd-0.5/0.8Ca alloys after the aging treatment in SBF. It can be clearly found that the Mg-6Zn-0.5Nd-0.5Ca alloy had the lower corrosion rate than Mg-6Zn-0.5Nd-0.8Ca alloy. With the extension of corrosion time, the corrosion rate dropped in an approach exponential form for two alloys. The corrosion rates of Mg-6Zn-0.5Nd-0.5/0.8Ca alloys were 0.45 and 0.69 mg/cm^2^·h after the corrosion for one hour, and reached the relatively stable values of 0.05 and 0.16 mg/cm^2^·h after the corrosion for 192 h.

The corrosion rates of the two groups of alloys were shown in Figure 8b. The corrosion rate of Mg-6Zn-0.5Nd-0.5Ca alloy was 0.82 mm/a, which was significantly lower than that of Mg-6Zn-0.5Nd-0.8Ca alloy (2.98 mm/a). The change in pH value mainly reflected the change in the concentration of H^+^ or OH^−^ in the solution. The changing rate of pH value also suggested the corrosion rate. Figure 8c showed the change in pH value during corrosion of Mg-6Zn-0.5Nd-0.5/0.8Ca alloy immersed in SBF. It was can be seen that the pH values of the Mg-6Zn-0.5Nd-0.5/0.8Ca alloys increased rapidly within 72 h due to the release of OH^−^. The pH value of Mg-6Zn-0.5Nd-0.5/0.8Ca alloys successively reached 8.04 and 8.26 after 72 h. With the further extension of time, the change of pH value increased slowly, indicating the corrosion rate dropped slightly. This result was consistent with the results obtained from the weightless corrosion. As shown in Figure 8d, the self-corrosion potential of Mg-6Zn-0.5Nd-0.5Ca alloy was −1.57 V, and the self-corrosion current density was 26.72 μA/cm^2^. The self-corrosion potential of Mg-6Zn-0.5Nd-0.8Ca alloy was −1.63 V, and the self-corrosion current density was 24.87 μA/cm^2^. Therefore, the Mg-6Zn-0.5Nd-0.5Ca alloy had the better corrosion resistance.

Fourier transform infrared spectroscopy (FTIR) was used to explore the composition of corrosion products and the results were shown in Figure 9. The absorption peak at 3693 cm^−1^ was formed by the stretching vibration of Mg-OH. The absorption peak at 2921 cm^−1^ was the HPO_4_^2−^ group. The absorption peaks of 1449 and 862 cm^−1^ were CO_3_^2−^ groups. The absorption peak positions of 1167, 1053, and 576 cm^−1^ were PO_4_^3−^ groups. The substance with the absorption peaks at 3450 and 1645 cm^−1^ was H_2_O.

In order to further investigate the corrosion process of Mg-6Zn-0.5Nd-0.5Ca alloy, the corrosion products were analyzed. Typically, some corrosion products were adsorbed on the surface of magnesium alloys, and others deposited at the bottom of beaker after the immersion. Once the magnesium alloy was immersed in the corrosive solution, the corrosion products would form on the surface of alloy. After the immersion for one hour, the honeycombed, short rod and spherical corrosion products were formed on the surface of magnesium alloy. The honeycombed corrosion product with high porosity was regarded as Mg(OH)_2_, according to the literature [18]. However, the compositions of two other corrosion products remain unclear. Therefore, the EDS analyses of them were conducted.

Figure 10 showed the morphologies and EDS spectra of the two main corrosion products, and the results of element content analysis were listed in Table 4. As shown in Figure 10, the spherical corrosion products had more compact structure than the net-like corrosion products. Hence, the spherical corrosion products were conducive to improving the corrosion resistance. The net-like corrosion products had the diameter of ca. 0.4 μm and evenly distributed on the surface of alloy (Figure 10a). The spherical corrosion products had the diameter of ca. 1.3 μm, and tightly bonded with each other (Figure 10b). Based on the FTIR and EDS results, it can be speculated that spherical and net-like corrosion products included Mg(OH)_2_, MgCO_3_, Ca_3_(PO_4_)_2_, Ca(H_2_PO_4_)_2_ and CaCO_3_. In addition, it can be clearly found from Table 4 that the spherical corrosion products had more O element of 56.24 at% and less C (27.36 at%) and P (5.59 at%) elements than the net-like corrosion products (49.78 at% O, 29.45 at% C and 7.36 at%). Furthermore, the spherical corrosion products had the higher ratio of Mg to Ca content (1.76) than the short net-like corrosion products (1.46). Thus, the results indicated that the spherical corrosion products had more Mg(OH)_2_ phase and more compact structure than the net-like corrosion products, resulting in the improvement of the corrosion resistance [33]. Besides, more Mg(OH)_2_ phase formed and deposited on the surface of the alloy with the extension of corrosion time, resulting the decrease in the corrosion rate. This result was consistent well with the results obtained from Figure 8.

Based on the above analysis results, the chemical reactions of Mg-Zn-Nd-Ca alloy in SBF solution were rather complicated, and some possible reactions were as follows:Electrochemical reaction process:Anode process: Mg → Mg^2+^ + 2e^−^Cathode process: 2H^+^ + 2e^−^ → H_2_↑Chemical reaction process:
Mg → Mg^2+^ + 2e^−^
2H_2_O + 2e^−^→ 2OH^−^ + H_2_↑
Mg^2+^ + 2OH^−^ → Mg(OH)_2_↓
HCO_3_^−^ → H^+^ + CO_3_^2−^
Mg^2+^ + CO_3_^2−^ → MgCO_3_↓
Ca^2+^ + CO_3_^2−^ → CaCO_3_↓
Ca^2+^ + HPO_4_^2−^ + 2H_2_O → CaHPO_4_·2H_2_O↓
3Ca^2+^ + 2PO_4_^3−^ + nH_2_O → Ca_3_(PO_4_)_2_·nH_2_O↓
Mg(OH)_2_ + 2Cl^−^ → MgCl_2_ + 2OH^−^

Figure 11 showed a schematic diagram of the corrosion process of Mg-6Zn-0.5Nd-xCa alloy. The self-corrosion potential of α-Mg matrix was −2.37 V, and the secondary phase particles in the alloy had a higher self-corrosion potential. Hence, the galvanic corrosion occurred on the surface of alloy under the drive of the difference in self-corrosion potential, once the alloy was immersed in the SBF solution.

As the corrosion time prolonged, the massive corrosion products almost cover the entire surface, forming a compact layer, which hindered the deep development of the pitting holes and improve the corrosion resistance of the alloy. It should be noted that the corrosion rate increased chloride under the existence of chloride ion. This was because that the chloride ion had a small radius and high permeability, and was easily adsorbed on the surface of the alloy. In addition, the higher the concentration of chloride ion in solution was, the more MgCl_2_ phase was formed through the collision of chloride ion and Mg atoms on the surface of alloy. This result further led to that the Mg(OH)_2_ layer was destroyed and the corrosion rate increased. However, the compact Mg(OH)_2_ layer can significantly improve the corrosion resistance via reducing the contact area between chloride ion and alloy.

## 4. Conclusions

(1)When the Mg-6Zn-0.5Nd-0.5/0.8Ca alloys were extruded at medium temperature, the surface of the extruded alloys was smooth. After the aging treatment, the grains were refined and the fine secondary phase particles distributed uniformly on the α-Mg matrix.(2)Mg-6Zn-0.5Nd-0.8Ca alloy had better mechanical properties than Mg-6Zn-0.5Nd-0.5Ca alloy, while its corrosion resistance became lower. MgZn_2_ nanoparticles and dislocations played a positive role in improving the strength of the alloy.(3)The corrosion rates of Mg-6Zn-0.5Nd-0.5/0.8Ca alloys were 0.82 and 2.98 mm/a, respectively. Although the existence of chloride ions accelerated the corrosion reaction, the compact layer constituted of the corrosion products can effectively hinder the permeation of Cl^−^ and improve the corrosion resistance of the Mg-6Zn-0.5Nd-xCa alloys.

## Figures and Tables

**Figure 1 materials-12-01049-f001:**
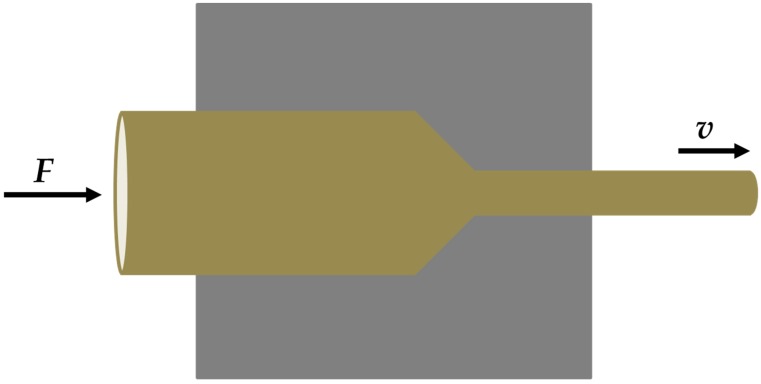
Schematic diagram of the extrusion process.

**Figure 2 materials-12-01049-f002:**
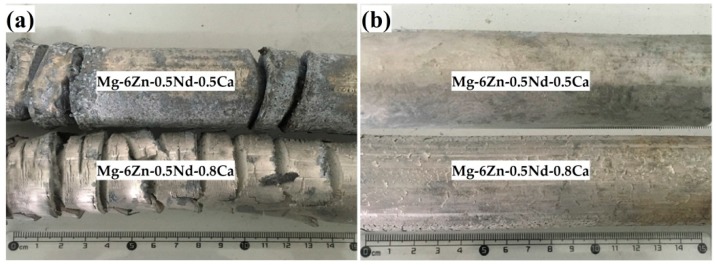
Appearance of the as-extruded Mg-6Zn-0.5Nd-0.5/0.8Ca alloys: (**a**) at high extrusion temperature; and (**b**) at medium extrusion temperature.

**Figure 3 materials-12-01049-f003:**
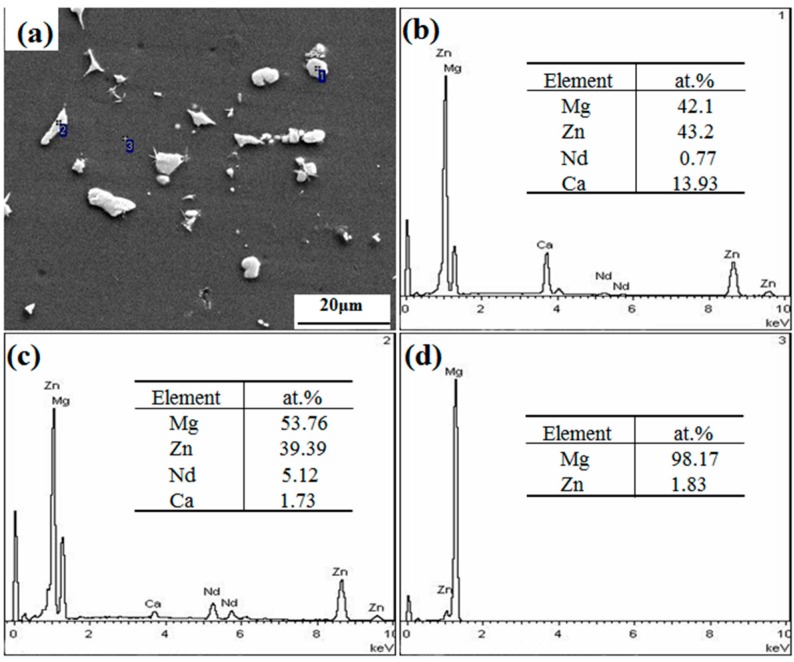
SEM image and EDS spectra of the extruded Mg-6Zn-0.5Nd-0.5Ca alloy. (**a**) SEM image of extruded Mg-6Zn-0.5Nd-0.5Ca alloy; (**b**) EDS spectra and element content of Point 1; (**c**) EDS spectra and element content of Point 2; (**d**) EDS spectra and element content of Point 3.

**Figure 4 materials-12-01049-f004:**
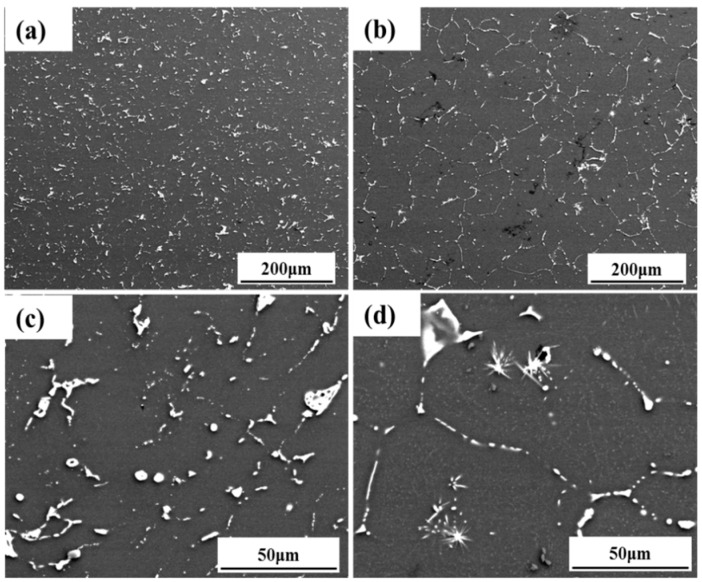
SEM images perpendicular to the extrusion direction of the extruded Mg-6Zn-0.5Nd-0.5/0.8Ca alloys. (**a**) is Mg-6Zn-0.5Nd-0.5Ca alloy; (**b**) is Mg-6Zn-0.5Nd-0.8Ca alloy; and (**c**,**d**) are the enlargement of (**a**,**b**), respectively.

**Figure 5 materials-12-01049-f005:**
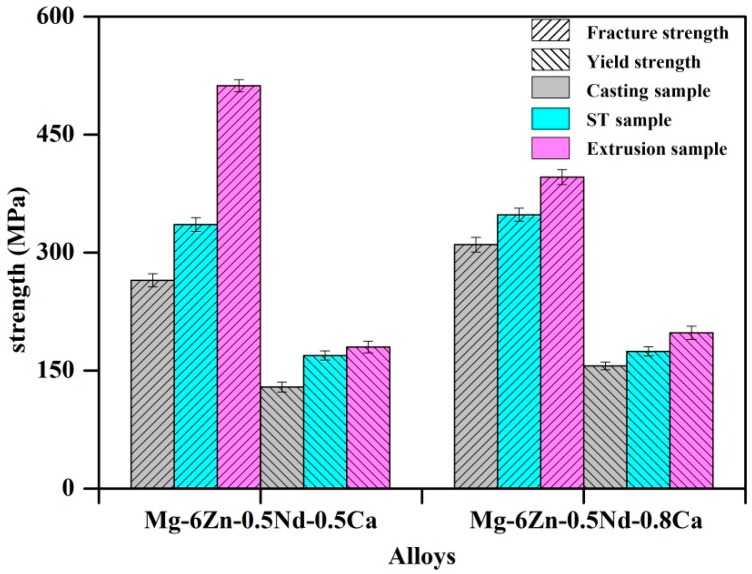
Compression mechanical properties of Mg–Zn–Nd–Ca alloys.

**Figure 6 materials-12-01049-f006:**
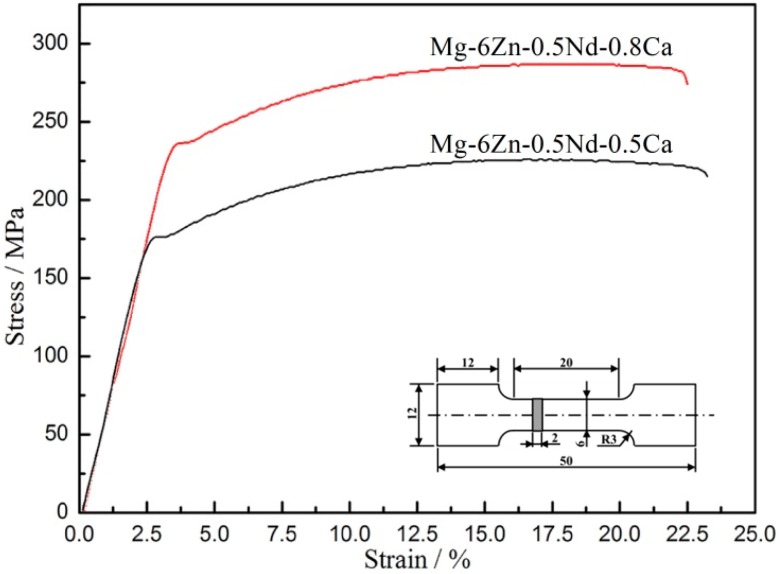
Stress-strain curves of the extruded Mg-6Zn-0.5Nd-0.5/0.8Ca alloys after the aging treatment and the dimensions of tensile specimen (unit: mm).

**Figure 7 materials-12-01049-f007:**
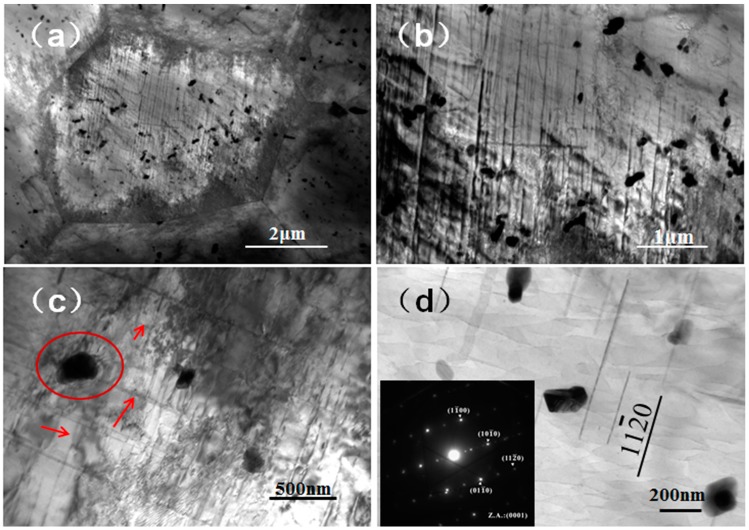
TEM analysis of Mg-6Zn-0.5Nd-0.8Ca alloy extruded at medium temperature. (**a**) is a bright field phase of the grain. (**b**) is an enlarged view of (**a**). (**c**) is a bright-field micrograph of the dislocation and the precipitated phase. (**d**) is the precipitated phases of small particles and selected area electron diffraction pattern.

**Figure 8 materials-12-01049-f008:**
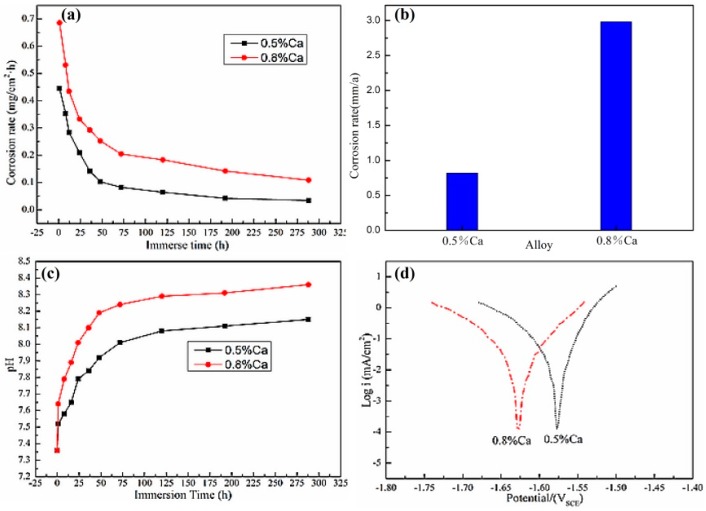
Corrosion properties of the extruded Mg-6Zn-0.5Nd-0.5/0.8Ca alloys after aging treatment. (**a**) shows the corrosion rate of alloys at different etching times; (**b**) depicts the average corrosion rate of alloys; (**c**) displays the curves of pH value vs. immersion time; and (**d**) represents the polarization curves of alloys in SBF.

**Figure 9 materials-12-01049-f009:**
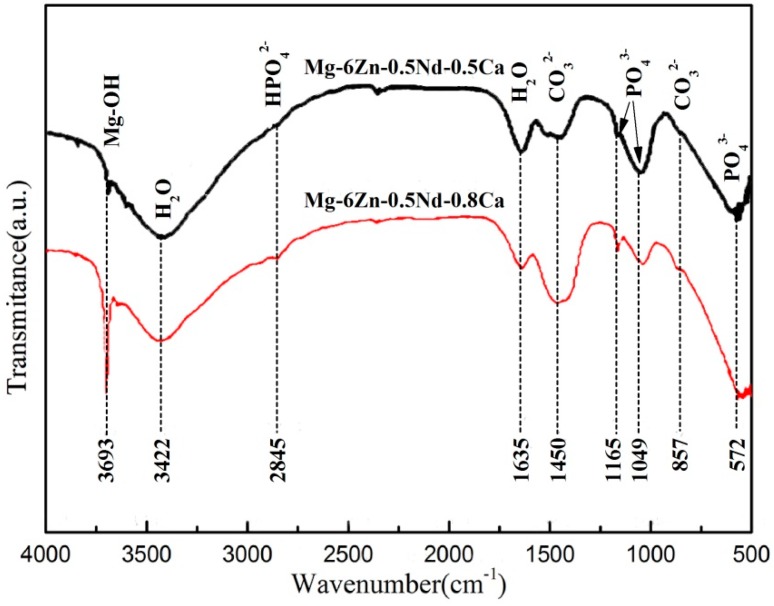
FTIR spectra of corrosion products of the extruded Mg-6Zn-0.5Nd-0.5/0.8Ca alloys after the aging treatment.

**Figure 10 materials-12-01049-f010:**
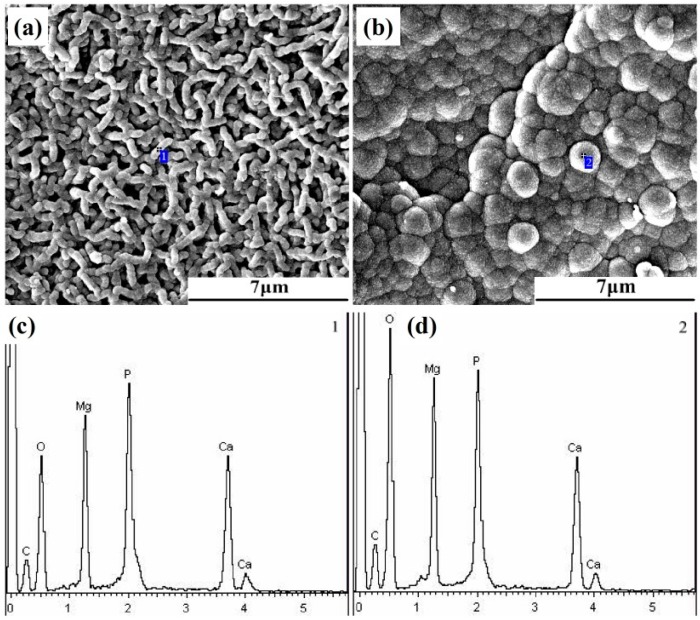
SEM images of corrosion products of the extruded Mg-6Zn-0.5Nd-0.5Ca alloy after aging treatment. Both (**a**,**b**) are the microscopic morphology of corrosion products. (**c**,**d**) are the EDS results of the corrosion products (**a**,**b**), respectively. (**c**,**d**) are EDS spectra at Point 1 and 2 in (**a**,**b**), respectively

**Figure 11 materials-12-01049-f011:**
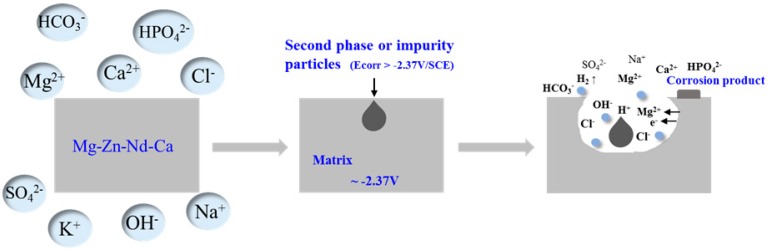
Schematic diagram of the corrosion process of Mg-6Zn-0.5Nd-xCa alloys.

**Table 1 materials-12-01049-t001:** Chemical composition of the extruded alloys, wt%.

Alloy	Zn	Nd	Ca	Al	Fe	Mg
Mg-6Zn-0.5Nd-0.5Ca	6.066	0.475	0.517	0.034	0.037	Bal.
Mg-6Zn-0.5Nd-0.8Ca	5.865	0.522	0.783	0.045	0.028	Bal.

**Table 2 materials-12-01049-t002:** The concentrations of various ions in the SBF.

Ion	Ion Concentration (mM)	
	Blood Plasma	SBF
Na^+^	142.0	142.0
K^+^	5.0	5.0
Mg^2+^	1.5	1.5
Ca^2+^	2.5	2.5
Cl^−^	103.0	103.0
HCO_3_^−^	27.0	27.0
HPO_4_^2−^	1.0	1.0
SO_4_^2−^	0.5	0.5
pH	7.2–7.4	7.3

**Table 3 materials-12-01049-t003:** Mechanical properties of the extruded Mg-6Zn-0.5Nd-0.5/0.8Ca alloys after the aging treatment.

Alloy	Mg-6Zn-0.5Nd-0.5Ca	Mg-6Zn-0.5Nd-0.8Ca
σ_e_ (MPa)	168.5	229.7
σ_s_ (MPa)	173.3	235.8
σ_b_ (MPa)	222.5	287.2
δ (%)	20.2	18.4

**Table 4 materials-12-01049-t004:** EDS results of the corrosion products on the surface of Mg-6Zn-0.5Nd-0.5Ca alloy, at%.

Point	C	O	Mg	P	Ca
Point 1	29.45	49.78	7.96	7.36	5.45
Point 2	27.36	56.24	6.89	5.59	3.92

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
