# Peer review of "Effects of Extrusion on Mechanical and Corrosion Resistance Properties of Biomedical Mg-Zn-Nd-xCa Alloys"

_materials, 2019, doi:10.3390/ma12071049_

Reviewer 1 Report

Dear authors,

Manuscript ID materials- 463885 “Effect of extrusion on mechanical and corrosion resistance properties of Mg-Zn-Nd-xCa alloy”, I recommend publishing this paper after minor revisions according by following suggestions.

The main comments:

Please, fill and correct your article according to several suggestions

Please check the References section

Improve quality of pictures

Add instrumental method - operating parameters, precision, accuracy and uncertainties of methods, QA/QC procedure

Can you have quantified the amount of corrosion products Mg(OH)2, MgCO3, Ca3(PO4)2, CaHPO4 and CaCO3.

Can you add others authors from Europe, not only from Asia

for example

N. Martynenko, E. Lukyanova, V. Serebryany, D. Prosvirnin, V. Terentiev, G. Raab, S.       Dobatkin and Y. Estrin, Effect  of equal channel angular pressing on structure, texture, mechanical and in-service properties of a biodegradable magnesium alloy, Materials Letters, 10.1016/j.matlet.2018.12.024, 238,         (218-221), (2019).

Kamineni  Pitcheswara Rao, Yellapregada Venkata Rama Krishna Prasad, Chalasani Dharmendra, Kalidass Suresh, Norbert Hort and Hajo Dieringa, Review on Hot Working Behavior and Strength of Calcium‐Containing Magnesium Alloys, Advanced Engineering Materials, 20,         9, (2018).

DOLEŽAL, P.; ZAPLETAL, J.; FINTOVÁ, S.; TROJANOVÁ, Z.; GREGER, M.; ROUPCOVÁ, P.; PODRÁBSKÝ, T. Influence of Processing Techniques on Microstructure and Mechanical Properties of a Biodegradable Mg-3Zn- 2Ca Alloy. Materials, 2016, 9 (11), 880I recommend accept manuscript after minor revision.  

Sincerely

Author Response

Thank you very much for your comments. We have replied to your comments in the attachment.

Reviewer 2 Report

1.      Describe, please, how the samples for the microstructural characterization were prepared (Fig. 1.) Sandpapers, etching solution etc.

2.      Add the information concerning the TEM-images: samples preparation, what type of mesh was used.

3.      Add detailed information on how the data presented in Fig 4a was obtained.

4.      Revise the text of the manuscript to insert the spaces: 287.15MPa, 0.82mm/a etc, the same for Fig. 4 d.

5.      Provide some additional proofreading of the manuscript and revise some sentences, for example "the hydrogen ions generating electrons to generate a large amount of hydrogen", " the corrosion products gradually accumulated on the surface of the alloy, and the surface of corrosion products was mostly compact".

6.      The calculated values of corrosion current are high enough for Mg alloys, and calculated values of corrosion rates are low. It would be useful to provide comparison of obtained values with existing literature data, for example doi:10.3390/met8040238.

7.      Provide the optical images of corroded surfaces of Mg alloys after different immersion times.

8.      The major point is that there is no discussion concerning the reasons Why the corrosion rates differ for more than three times.

Author Response

(The authors gave the same response as above.)

Reviewer 3 Report

The article title and Keywords should include a particular word such as ‘Biodegradable’ or ‘Biomedical’. The present title does not express the biodegradable Mg alloy.

The graph line colors should be unified among figures 2, 4, 5. 

The extrusion process should be clearly described with the dimensions and the process rate, preferably with an illustration. Also the volume of SBF solution should be described in the evaluation of corrosion resistance.

Because this article has focused on ‘effect of extrusion’, this article should involve the results of the control materials without extrusion process, i.e. the specimens made from the cast ingot after solution treatment. Regarding the corrosion rate, it is preferable to compare with pure Mg or commercial Mg alloys.

Regarding SEM images in Figure 1, the author suddenly stated the grain boundary and the secondary phase in the main text. However, the author should kindly explain the bright parts in the SEM images, e.g. precipitated solid solution with different crystal phase or Intergranular compounds / Intermetallic compounds. Preferably together with their chemical composition by EDX analysis.And the summary should add to the figure caption.

It is well known that Mg, Zn, Ca are included in human body, however, the biocompatibility (the acute and chronic toxicity) of Nd is unknown. The author should discuss Nd as the alloy element for biodegradable implant.

Author Response

Thank you very much for your comments. We have replied to your comments in the attachment.

Round  2

Reviewer 3 Report

The article is well improved. However, even if the author already reported the previous work on the mechanical and corrosion properties of as-cast and heat-treated Mg-Zn-Nd-xCa alloys. In order to enhance the value of this article on the effect of extrusion, it is strongly recommended to refer the previous work in the section of Mechanical Properties and Strengthening Mechanism.

Author Response

Thank you very much for your comments. We have replied to your comments in the attachment.

Materials EISSN 1996-1944 Published by MDPI AG, Basel, Switzerland RSS E-Mail Table of Contents Alert
Back to Top